# Adaptive Response and Transcriptomic Analysis of Flax (*Linum usitatissimum* L.) Seedlings to Salt Stress

**DOI:** 10.3390/genes13101904

**Published:** 2022-10-20

**Authors:** Yuandong Li, Jiao Chen, Xiao Li, Haixia Jiang, Dongliang Guo, Fang Xie, Zeyang Zhang, Liqiong Xie

**Affiliations:** Xinjiang Key Laboratory of Biological Resources and Genetic Engineering, College of Life Science and Technology, Xinjiang University, Urumqi 830046, China

**Keywords:** flax, salt stress, adaptive growth, osmoregulation, antioxidant properties, transcriptome analysis

## Abstract

Soil salinity constrains agricultural development in arid regions. Flax is an economically important crop in many countries, and screening or breeding salinity-resistant flax cultivars is necessary. Based on the previous screening of flaxseed cultivars C71 (salt-sensitive) and C116 (salt-tolerant) as test materials, flax seedlings stressed with different concentrations of NaCl (0, 100, 150, 200, and 250 mmol/L) for 21 days were used to investigate the effects of salt stress on the growth characteristics, osmotic regulators, and antioxidant capacity of these flax seedlings and to reveal the adaptive responses of flax seedlings to salt stress. The results showed that plant height and root length of flax were inhibited, with C116 showing lower growth than C71. The concentrations of osmotic adjustment substances such as soluble sugars, soluble proteins, and proline were higher in the resistant material, C116, than in the sensitive material, C71, under different concentrations of salt stress. Consistently, C116 showed a better rapid scavenging ability for reactive oxygen species (ROS) and maintained higher activities of antioxidant enzymes to balance salt injury stress by inhibiting growth under salt stress. A transcriptome analysis of flax revealed that genes related to defense and senescence were significantly upregulated, and genes related to the growth and development processes were significantly downregulated under salt stress. Our results indicated that one of the important adaptations to tolerance to high salt stress is complex physiological remediation by rapidly promoting transcriptional regulation in flax.

## 1. Introduction

Salinity is a major abiotic stress factor that reduces plant growth and productivity [1,2]. More than 200 million hectares of land in East and Central Asia (approximately 22% of the world’s total saline–alkali land area) are affected by salt build-up, and the salinization of China’s arable land is estimated at 7.6 million hectares [3,4]. Flax (*L. usitatissimum* L.) is an important oil crop widely cultivated in many Asian countries [5,6]. The Food and Agriculture Organization of the United Nations (FAO) reported that Kazakhstan and Russia are the newest top five flaxseed producers, just as China and India were [7]. However, water shortages in these areas increase soil salinity and negatively affect flax yield and quality [4,8]. The increased number of heat waves and droughts across the globe has exacerbated the formation of saline–alkali soils and the current food crisis [9,10]. Therefore, breeding and screening salinity-tolerant flax varieties is a beneficial way to enhance crop production and utilize salinity in soil and water.

Salt stress causes ion imbalance, osmotic derangement, and the accumulation of toxic substances, especially reactive oxygen species (ROS), in plants [11,12]. Plants balance the high density of salt in cells by remodeling relevant transcriptional networks and generating specific physiological responses [13,14]. Compatible osmolytes, including charged metabolites (e.g., proline and glycine betaine), polyols (e.g., mannitol and sorbitol), sugars (e.g., sucrose and fructose), complex sugars (e.g., trehalose and raffinose), and ions (e.g., K^+^), are important for salt-induced osmotic regulation [15,16,17,18]. Previous studies have reported that plants can produce more proline and soluble sugars, whose cell concentrations are rapidly modified under salt or water stress [15,19]. In addition, salinity increases the ROS concentration, which has deleterious effects on plant physiology [20,21]. Catalase (CAT) and superoxide dismutase (SOD) are key proteins involved in ROS scavenging and are known as the first line of defense against ROS toxicity [22,23]. As a recently developed tool, transcriptomic analysis has contributed significantly to elucidating adverse stress responses [24,25,26]. To date, many studies of salt tolerance have been conducted on *Arabidopsis thaliana* [27,28], *Oryza sativa* [29,30], and other crops [31,32] using RNA-Seq technology. Physiological and transcriptomic analyses of salt resistance help us understand plant adaptation mechanisms in severe environments.

Since flax was first cultivated in 9000 BC in Egypt, the crop has shown extensive variation and is distributed worldwide [5,33]. Rich biodiversity enables the species to exhibit changeable adaptations with different physiological responses and behavioral patterns to salt stress [34,35]. Many previous studies have evaluated salt-stress adaptive changes in flax populations during germination [36,37]. Indicators such as germination rate, root length, and shoot length effectively reveal the behavioral adaptation of flax to salt stress during the germination period [38,39]. Abido et al. [40] found that Egyptian flax cultivars showed wide natural variation in adaptation to salinity, and the Giza11 cultivar exhibited a higher germination rate and speed at high salinity levels compared with other cultivars. Li et al. [41] used GWAS to mine an SNP locus associated with salt stress during flax germination, and a haplotype analysis of alleles based on this SNP showed that flax accessions containing the GG haplotype were more salt-adapted than those containing the AA haplotype. Some flax cultivars are salt-tolerant during germination but salt-sensitive during the vegetative phase [42]. The seedling stage is an important period for crop growth and yield [8,38]; however, adaptive changes in flax seedlings under salt stress have been less studied. In particular, no transcriptional profiles other than Wu’s limited study [43], in which only 42.92% of RNA reads were mapped to the reference genome, have shown differentially expressed genes (DEGs) to analyze the difference in adaptability between flax cultivars under salt stress. Therefore, it is necessary to explore the adaptive responses of flax seedlings to salt stress.

To breed flax cultivars most suitable for growth in northwest China, we collected thousands of flax cultivars from around the world and constructed a widely representative core population by analyzing their genetic diversity [33]. The salt tolerance of 200 flax accessions in the germination stage was evaluated using the *D*-value, and the salt-tolerant material C116 and the salt-sensitive material C71 were analyzed [41]. In this study, we investigated the differences in osmoregulator concentration and antioxidant capacity between resistant and susceptible materials at the seedling stage. We combined this with transcriptomic data to elucidate the relationships between physiological responses and gene expression under salt resistance in flax. Our results provide insights into the salt stress adaptation mechanisms of flax seedlings with the hope of accelerating the development of salt-tolerant varieties.

## 2. Materials and Methods

### 2.1. Plant Materials and Salt Treatment

Seeds of two flax cultivars, salt-tolerant (C116) and salt-susceptible (C71), were selected based on salt-tolerance indices [41]. The seeds were collected from the Flax Germplasm Repository, College of Life Sciences and Technology, Xinjiang University, Xinjiang, China. Flax seeds were germinated in a moist environment for 7 days. After germination, the seedlings were transferred to 1/2 Hoagland nutrient solution for 2 weeks before NaCl treatment. Twenty-one-day-old seedlings were treated with 0, 100, 150, 200, and 250 mmol/L NaCl for seven days. Phenotype data, including survival rate, plant height, root length, seedling dry weight, and seedling fresh weight, were recorded daily. After salt treatment, fresh leaves and roots were collected on days 1 and 5. Seedlings were collected after 3 days of salt stress and kept at −40 °C for all physiological experiments. Each treatment was performed in triplicate.

### 2.2. Data Measurement for the Hydroponic Experiment

#### 2.2.1. Seedling Parameters

Thirty-six flax seedlings of each of the two cultivars were planted in the same hydroponic pot and recorded as replicates. After 5 days of salt stress, the surviving plants were photographed and counted manually. The experiments were performed in triplicate.Seedlings for root and shoot growth assays were planted in punched 96-well PCR plates. For the assessment of plant height and root length, at least 15 seedlings were measured from three plates in each experiment, and three independent experiments were performed. The f RPH (relative plant height) and RRL (relative root length) formulas were used to calculate the relative plant height and relative root length: RPH = (PT − PC)/PC × 100%, RRL = (RT − RC)/RC × 100%. PT, plant height after 5 days of salt stress; PC, plant height before salt stress; RT, root length after 5 days of salt stress; RC, root length before salt stress.Whole flax seedlings were selected, the external moisture was removed using absorbent paper, and the fresh weight (WF) was recorded. The seedlings were then killed at 105 °C for 1 h, followed by drying at 80 °C for 8 h, and the dry weight (WD) was recorded. The RWC formula was used to calculate the relative water content. RWC = (TWF − TWD)/(CWF − CWD) ×100%. TWF, WF of seedlings after 5 days of salt stress; CWF, WF of seedlings before salt stress; TWD, WD of seedlings after 5 days of salt stress; CWD, WD of seedlings before salt stress.

#### 2.2.2. Chemical Molecule Analysis

The proline content in this study was quantified using a proline assay kit (A107-1-1; Nanjing Jiancheng Bioengineering Institute, Nanjing, China). Seedling roots (0.1 g) were ground in liquid nitrogen, and the proline concentration was determined according to the manufacturer’s protocol.The soluble sugar content was quantified using a plant soluble sugar content test kit (A107-1-1; Nanjing Jiancheng Bioengineering Institute, Nanjing, China). Seedling roots (0.1 g) were ground in liquid nitrogen, and the concentration of soluble sugars was determined according to the manufacturer’s protocol.The soluble protein determination was conducted using a previously reported method [44]. The seedling roots were used in this study. Soluble proteins were quantified using a spectrophotometer (595 nm) and expressed on a fresh-matter basis. Bovine serum albumin was used as the standard.The SOD activity was determined using a total superoxide dismutase assay kit (A007-1-1; Nanjing Jiancheng Bioengineering Institute, Nanjing, China). One unit of SOD activity was defined as the amount of enzyme required for 1 g of tissue in 1 mL of a reaction mixture SOD inhibition ratio of 50% as monitored at 550 nm. The SOD activity was expressed as U/g fresh weight.CAT activity was determined using a catalase assay kit (A007-1-1; Nanjing Jiancheng Bioengineering Institute, Nanjing, China). One unit of CAT activity was defined as 1 mg tissue proteins consuming 1 µmol H_2_O_2_ at 405 nm for 1 s. The CAT activity was expressed as U/mg of protein.

#### 2.2.3. Reactive Oxygen Species (ROS) Staining

Nitro blue tetrazolium chloride (NBT, Sigma-Aldrich, St. Louis, MO, USA, N6876) and 3,3′-Diaminobenzidine (DAB, Sigma-Aldrich, St. Louis, MO, USA, D8001) were freshly prepared by dissolving them in phosphate buffer (pH 3.8) at a final concentration of 1 mg/mL. We used young leaves and roots after 3 days of salt stress according to the method outlined in a previous study [45]. The two staining protocols were similar. Briefly, the leaves and roots were incubated in the staining solution for 4 h at 37 °C, the solution was removed, and the leaves were boiled for 30 min in an ethanol solution. After cooling, leaves and roots were placed under a stereomicroscope and photographed using a scanner.

### 2.3. RNA-Seq Data Analysis

RNA-seq data analysis methods were referenced from Qi et al. [46], with some modifications. Briefly, we downloaded the raw transcriptome data, PRJNA860005, from NCBI using the same tools. Next, we used fastp for quality control, filtered the raw data, and obtained the Q20, Q30, and clean data. The similarity between the data and samples can be obtained by comparing them with the published schedules. The paired-end clean reads were then aligned to the “*L. usitatissimum* cv. longya10” flax genome using Hisat2 v2.1.0 to generate mapping results. Subsequently, featureCounts v1.6.0 was used to count the read numbers mapped to each gene [47]. The transcripts per kilobase of million mapped reads (TPM) for each gene were calculated based on the length of the gene and the read count mapped to this gene. The limma program was used to correct the results obtained in the previous step and obtain the TPM and trimmed mean of M values (TMM) two-fold normalized matrices for gene expression data [48].

### 2.4. Screening of DEGs and Enrichment Analysis

Differential expression analysis was performed using the DESeq2 R package (1.20.0) [49]. The *p*-values were adjusted using the Benjamini–Hochberg method to control the false discovery rate. A corrected *p*-value of 0.05 and log_2_ (fold change) of 2 were set as the threshold for significantly differential expression.

For rapid GO and KEGG enrichment analyses, it was necessary to construct a flax-specific Orgdb annotation package. Fast annotation of flax protein sequences was performed using EGGNOG-Mapper (http://eggnog-mapper.embl.de/, accessed on 16 October 2022) to obtain annotation files, followed by the AnnotationForge package to build Orgdb packages exclusive to flax. The Orgdb annotation package can assist clusterProfiler in performing GO and KEGG enrichment analyses quickly. For gene expression profiling analysis, functional assignments were mapped to GO terms [50]. Significantly enriched pathways were identified according to *p*-values and enrichment factors [51].

### 2.5. Quantitative Real-Time PCR Validation

DEGs found in the transcriptome sequencing analysis were verified using quantitative real-time PCR (qRT-PCR) to support the findings of the RNA-Seq study of gene expression. Six genes (*Lus10012145*, *Lus10031258*, *Lus10029081*, *Lus10002083*, *Lus10005114*, and *Lus10041534*) were randomly selected to analyze expression levels in flax treated with NaCl solution exposure times of 24 h and 72 h using *L. usitatissimum ACT1* (GenBank accession no. AY857865) as the internal reference gene. cDNA was synthesized using 5X All-In-One RT MasterMix (with the AccuRT Genomic DNA Removal Kit). Primer sets were designed using Primer Premier v6.25 (http://www.premierbiosoft.com/crm/jsp/com/pbi/crm/clientside/ProductList.jsp, accessed on 16 October 2022) and are listed in Appendix A. qRT-PCR assays were performed using an ABI PRISM 7500 real-time PCR system (USA). A standard curve was used to estimate mRNA expression levels using critical thresholds. Data for each sample were corrected for loading with *LusAct1* as an internal reference gene using the 2^−ΔΔCt^ method [52]. The PCR program was as follows: 95 °C for 3 min, followed by 40 cycles of 95 °C for 15 s and 60 °C for 60 s. All samples were tested in triplicate.

### 2.6. Statistical Analysis

Statistical analysis of the corresponding experimental data was conducted using R version 4.0.4 and SPSS Version 20.0. The repeatability of RNA-seq samples was calculated using Pearson correlation coefficients (*R*^2^) based on the TMM value in R. Phenotypes and physiological data were analyzed using SPSS. Differences between survival rates were statistically analyzed using an unpaired two-tailed Student’s *t*-test. The mean values of the RWC, SOD, and CAT data were used to match the corresponding regression models and coefficients of determination. Other data were subjected to a one-way analysis of variance (ANOVA). Duncan’s new multiple range test (MRT) was applied when one-way ANOVA revealed significant differences. The average ± SD was used for phenotypic and physiological data analyses.

## 3. Results

### 3.1. Changes in the Growth of Flax under Salt Stress

The results showed that salt stress reduced the growth of both cultivars (Appendix A). The effect of increasing salt differed between the cultivars. When treated with 0, 100, 150, 200, or 250 mmol/L NaCl, the most obvious phenotypic differences were observed under 200 mmol/L NaCl stress (Figure 1A). After treatment with 200 mmol/L NaCl for half an hour, C116 showed head bending and lodging. The leaves of C71 began to wilt after 3 days of 200 mmol/L NaCl stress, and the seedlings completely wilted and died after 5 days. C116 had a significantly higher survival rate than C71 on day 7 (Figure 1B). This result was consistent with previous population screening results [41], which indicated that C116 plants were more tolerant to salt stress than C71 plants.

Plant height and root length are common indicators of plant growth and were used in this experiment. The 21 days after the germination of flax is considered a transitional period for seedlings from slow growth to fast growth [53]. We found that the plant height and root length of flax increased by an average of 1–2 cm per day during this period (Appendix A). This is a critical period for flax growth and development. We selected two fiber flax varieties; therefore, plant height can be used as an important factor to evaluate flax growth. With an increasing salt concentration, the RPH of C116 plants was significantly lower than that of C71 plants (Figure 1C). Salt stress inhibits the elongation of plant roots and whole-plant growth; therefore, root length changes can directly indicate the level of inhibition from salt stress [54]. Although the root growth rate of both cultivars showed a decreasing trend with increasing salt concentration, the roots of C116 were more sensitive to salt stress than those of C71. C116 showed short roots and slower root growth rates, whereas C71 roots grew faster (Figure 1D). The results for root and plant height showed that the growth of C116 was more severely inhibited under salt stress than that of C71.

### 3.2. Effect of Salt Stress on Osmoregulatory Substance Concentration

Excessive external salt concentrations can cause plant cells to lose water, resulting in physiological drought [35,55]. Both regression equations revealed a linear relationship between salt concentration and RWC, and the regression coefficient for C71 was lower than that of C116 (Figure 2A). When compared with the normal condition, the relative water content of C71 and C116 under 200 mmol/L NaCl stress decreased by an average of 7.21% and 4.21%, respectively. This indicated that C116 had a stronger water retention capacity than C71 under salt stress. The accumulation of osmoregulatory substances, such as proline, soluble sugars, and soluble proteins, can effectively balance intracellular and external water potentials to achieve water retention [56]. In the roots, the detection of relevant osmoregulatory substances showed that the concentrations of soluble sugars, soluble protein, and proline in C116 were significantly higher than those of C71 under 200 mmol/L NaCl stress. However, no difference was observed under lower salt concentrations (Figure 2B–D). The relative water content of C116 and C71 was almost the same under 200 mmol/L NaCl stress, and C116 mobilized more osmotic regulators in response to salt stress (Appendix A). We speculated that C116 improves the water retention capacity of flax by mobilizing osmoregulatory substances under high salt stress, thereby alleviating the corresponding osmotic pressure.

### 3.3. Effect of Salt Stress on the Accumulation and Scavenging of ROS

Excessive accumulation of ROS, especially H_2_O_2_ and O_2_^•−^, can disrupt cell membrane permeability and integrity and cellular compartmentalization [11]. We performed the NBT and DAB staining of flax roots and leaves for the preliminary determination of O_2_^•−^ and H_2_O_2_, respectively (Figure 3A–D). NBT and DAB staining showed that the staining of the treated roots and leaves of C71 was more intense than those of C116, indicating that C71 accumulated more O_2_^•−^ and H_2_O_2_ under salt stress, especially in leaves. Oxidative damage under salt stress can be alleviated by the expression of enzymatic and non-enzymatic free radical scavengers, with SOD being the main scavenging enzyme for O_2_^•−^ and CAT being the main scavenging enzyme for H_2_O_2_ [22]. The SOD and CAT activities showed nonlinear correlations with the salt concentration, and all tended to increase and then decrease with an increasing salt concentration (Figure 3E,F). The SOD activity of C71 was higher under low salt stress than that of C116, whereas the opposite was true under moderate-to-high salt stress. The SOD activity of C116 under 200 mmol/L NaCl stress was 2.3-fold higher than that of the control, whereas the SOD activity of C71 increased by only 0.43-fold. The overall CAT activity of C116 was higher than that of C71, and the differences in enzyme activity between the two cultivars under 200 mmol/L NaCl stress and 250 mmol/L NaCl stress were 39.4% and 48.2%, respectively (Appendix A). Both staining and enzyme activity assays indicated that the accumulation and scavenging of H_2_O_2_ may be one of the reasons for the difference in salt tolerance between the two cultivars.

### 3.4. Identification of DEGs under Salt Stress

To make the transcriptome results more representative, we chose “Longya10” (oil material) and “Fanni” (fiber material) for testing. Illumina paired sequencing technology was used to explore the relationship between NaCl stress-related DEGs, using three biological replicates per material. Mapping the rRNA depleted 582.3 million RNA-seq reads from the 12 samples against the improved flax reference genome [35] showed that 558.9 million reads (95.9%) were mapped in total, and 519.1 million (89.1%) were mapped uniquely (Figure 4A). The mean mapped reads per sample were 46.6 ± 4.5 million in total reads and 43.2 ± 4.2 million in uniquely mapped reads. Correlation analysis showed that the replicate samples had correlations above 0.8 (Appendix A). Gene expression profiles in response to NaCl stress were compared with those of the untreated control (water). In “Fanni” and “Longya10”, 7855 DEGs (3679 upregulated and 4176 downregulated) and 8054 DEGs (3970 upregulated and 4084 downregulated) were significantly differentially regulated in response to NaCl stress exposure, respectively (Figure 4B). A Venn diagram showed that 75.9% and 73.6% of the upregulated and downregulated genes, respectively, overlapped between the two varieties (Figure 4C). Most of these genes were associated with plant adversity stress using gene annotation (Appendix A). These differentially expressed genes play a crucial role in the adaptive response of flax under salt stress.

### 3.5. Functional Annotation of DEGs in Overlapping Regions

To further understand the functions of the DEGs, we performed a GO enrichment analysis. The results showed that 2905 upregulated and 3042 downregulated DEGs were significantly enriched in 126 and 81 biological processes (*Pad*j < 0.05), respectively (Figure 5A, Appendix A). The upregulated DEGs were mainly enriched in response to extracellular stimulus, the cellular response to abscisic acid stimulus, the abscisic acid-activated signaling pathway, anion transmembrane transport, toxic substances, aging, and plant organ senescence. The downregulated DEGs were mainly enriched in the generation of precursor metabolites and energy, photosynthesis, the response to cytokinin, the electron transport chain, the cellular response to auxin stimulus, the auxin-activated signaling pathway, and auxin transport. Therefore, the upregulated DEGs were mainly enriched in osmoregulatory pathways, defense responses, and aging, whereas the downregulated DEGs were mainly enriched in growth and development.

A KEGG pathway-based analysis was performed to further understand the biological functions and interaction pathways. KEGG enrichment analysis revealed that the upregulated and downregulated genes were associated with 316 (23 remarkable) and 319 (14 remarkable) KEGG pathways, respectively (Figure 5B, Appendix A). The upregulated DEGs were mainly involved in phenylpropanoid biosynthesis, plant-pathogen interaction, MAPK signaling pathway, oxidative phosphorylation, and the p53 signaling pathway. These pathways mainly control defense and stress responses in plants. In addition, downregulated genes were mainly associated with these pathways, including photosynthesis, purine metabolism, starch and sucrose metabolism, carbon fixation in photosynthetic organisms, porphyrin metabolism, methane metabolism, pentose phosphate pathway, and fatty acid elongation. These pathways primarily control the growth, differentiation, and metabolism of plant cells.

### 3.6. RNA-Seq Expression Validation with qRT-PCR

Six candidate DEGs were selected for a qRT-PCR analysis of resistant and susceptible materials to quantitatively assess the reliability and widespread repression of the transcriptome data. These results were consistent with the RNA-seq values obtained using each method, which differed by a log_2_-fold difference (Figure 6). Combining RNA-seq and qRT-PCR data to calculate their correlation, the results showed a positive correlation with the Pearson coefficient *R*^2^ = 0.889. Our results indicated that qRT-PCR expression profiles for the six selected DEGs were generally consistent with the RNA-seq results, thus demonstrating the feasibility and accuracy of the transcriptome analysis of NaCl stress in flax.

## 4. Discussion

Flax is an ancient crop that has generated many complex and sophisticated mechanisms to adapt to adversity during its evolution [5,53]. Deciphering the adaptive mechanisms of flax salt stress is beneficial for the scientific selection of salt-tolerant varieties and for alleviating food pressure. In this study, we attempted to reveal the adaptive responses of flax under salt stress by using the phenotypic changes and physiological responses of two flax cultivars with different salt tolerances under different salt concentrations. The results were verified by transcriptome analysis.

### 4.1. Growth Adjustment in Response to Salt Stress

Growth inhibition is an important means of flax resistance to salt stress pressure [34,40]. In this study, the biomass of both flax cultivars gradually decreased with increasing salt concentrations (Appendix A). With an increasing salt concentration, the plant height and root length of both cultivars were significantly reduced, and the relative reductions in plant height and root length of the salt-tolerant cultivar were greater than those of the salt-sensitive cultivar (Figure 1C,D). These results indicate that growth inhibition is one of the main responses to salt stress in flax. Furthermore, our transcriptomic data revealed that genes related to photosynthesis, auxin, and cytokinins were significantly downregulated (Figure 5A,B). It is well known that photosynthesis plays an important role in the formation of plants, and auxin and cytokinin are key regulators of growth and development [57,58]. These results suggest that sacrificing growth to improve salt tolerance is one of the strategies for flax adaptation under salt stress.

### 4.2. Osmotic Adjustment under Salt Stress

The rapid mobilization of the intracellular osmotic defense system can block the loss of plant cell water, thereby effectively reducing the impact of salt damage [1,55,59,60]. When plants are affected by salt stress, osmotic stress immediately decreases the cell expansion of root tips and young leaves and sharply diminishes stomatal conductance. Therefore, the plant cells lose water and die [10,14]. Many plants can develop compatible solutes to achieve stable osmotic pressure and to protect membranes and proteins from degradation by mediating osmotic adjustment [11,13,35]. Consistent with previous studies, our research showed that salt-induced accumulation was observed for most of the proline, soluble sugars, and soluble proteins in flax and that C116 accumulated more osmoregulatory substances than C71 at high salt concentrations (Figure 2B–D). C116 synthesized osmoregulatory substances at a significantly higher level than C71; in turn, it may obtain greater water-holding capacity and alleviate the lethal damage caused by high salt concentrations (Figure 2). Most strategies for improving water efficiency in plants focus on manipulating abscisic acid (ABA) signaling. ABA can activate SnRK2 under osmotic stress and is involved in regulating leaf starch hydrolysis through the SnRK2-AREB/ABF-BAM1/AMY3 signaling pathway, which is important for plant resistance to osmotic stress [55,58,61]. GO enrichment analysis revealed that 63 and 60 upregulated DEGs were enriched in the cellular response to abscisic acid stimulus and abscisic acid-activated signaling pathways, respectively (Appendix A). This demonstrates that the water-locking capacity of flax cells is one way to adapt to salinity.

### 4.3. ROS Homeostasis during Salinity Stress

ROS accumulation and scavenging are essential for the adaptive survival of flax under salt stress [62,63]. Salt stress induces the accumulation of reactive oxygen species in plant cells, leading to hyperoxia and the further production of toxic substances [21]. A transcriptome analysis identified 40, 50, and 63 upregulated DEGs enriched in response to reactive oxygen species, toxic substances, and the MAPK signaling pathway (classical detoxification signaling pathway), respectively. These pathways maintain ROS homeostasis [13,64]. It is well known that the excessive accumulation of ROS is fatal to plants, and the fundamental reason for this is the disruption of the ROS scavenging system, which leads to oxidative stress damage [11,65]. Some studies have demonstrated that H_2_O_2_ causes chloroplast clustering and thus affects photosynthetic machinery function under salt stress [14,22,66]. The exogenous application of CAT alleviates salt-induced harmful effects through H_2_O_2_ scavenging activity [65]. Similar to the previous results, both staining experiments and enzyme activity assays indicated that the accumulation and scavenging of hydrogen peroxide affected the salt tolerance of the two cultivars (Figure 3). Interestingly, we found 22 upregulated DEGs enriched in response to hydrogen peroxide (Appendix A). These results show that the accumulation and scavenging of reactive oxygen species are involved in the adaptive response of flax under salt stress, especially H_2_O_2_.

### 4.4. Transcriptome Analysis Using a High-Quality Reference Genome-Based Read Mapping

RNA-seq technology has become the main choice for detecting DEGs under different biological conditions [24,29]. The accurate, comprehensive, and precise interpretation of such high-throughput assays relies on well-characterized reference genomes and/or transcriptomes [43,46]. With the development of sequencing technology, the flax genome has been updated to chromosome-level genome assembly, and five high-quality flax genomes have been published [67,68,69]. Among these, Longya10 probably obtains a finer genome using Hi-C sequencing technology [68]. Fortunately, an annotation file for Longya10 was obtained. In this study, the mapping of RNA-seq reads against the improved reference genome showed an average mapping rate of 95.9%, among which the uniquely mapped reads averaged 89.1% (Figure 4A). The annotation file was used to simplify transcriptome analysis. Intriguingly, we captured some new DEGs that have not been annotated in previous studies; therefore, annotating these genes needs further validation (Appendix A).

## 5. Conclusions

The two flax cultivars studied, C116 and C71, were both affected by salt stress; however, differences in survival rates indicated that C116 was more adapted to salt stress than C71. An analysis of several behavioral and physiological parameters indicated that this difference in adaptation was due to nutrient regulation, water retention capacity, and ROS accumulation and scavenging. A transcriptome analysis revealed that defense- and senescence-related genes were significantly upregulated, and growth- and development-related genes were significantly downregulated under salt stress. These results provide a basis for understanding the adaptation mechanisms of flax seedlings under salt stress and have significance for the scientific selection of salt-tolerant flax germplasm.

## Figures and Tables

**Figure 1 genes-13-01904-f001:**
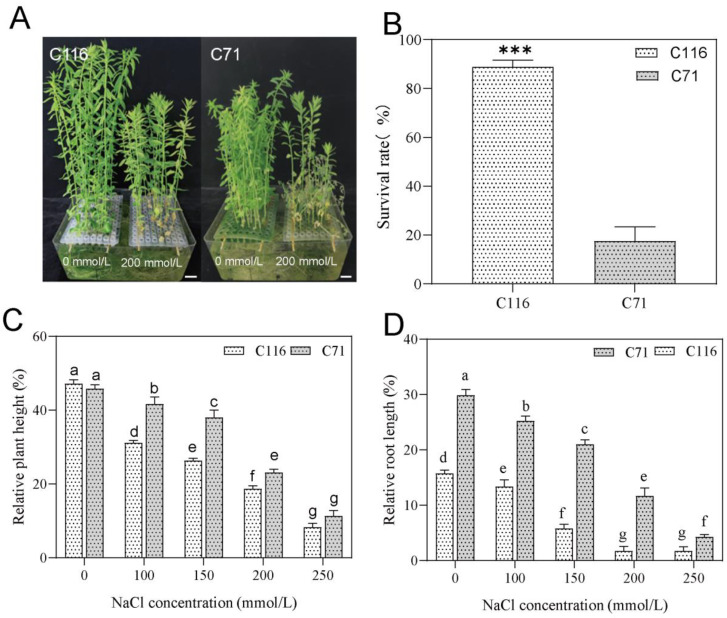
Statistical changes in the growth of flax under different salt concentrations. (**A**) Phenotypes (left: C116, right: C71; seedlings of the two concentrations were combined in water pots and photographed; scalebar = 1 cm). (**B**) Survival rates of the two species after 7 days of 200 mmol/L NaCl stress. Statistical differences between C116 and C71 survival rates were detected using Student’s *t*-test; *** *p* < 0.001. (**C**) Relative plant height and (**D**) relative root length under different salt concentration treatments. Error bars represent mean ± SD (*n* > 3), MRT; adjusted *p* < 0.001 is indicated by different lowercase letters.

**Figure 2 genes-13-01904-f002:**
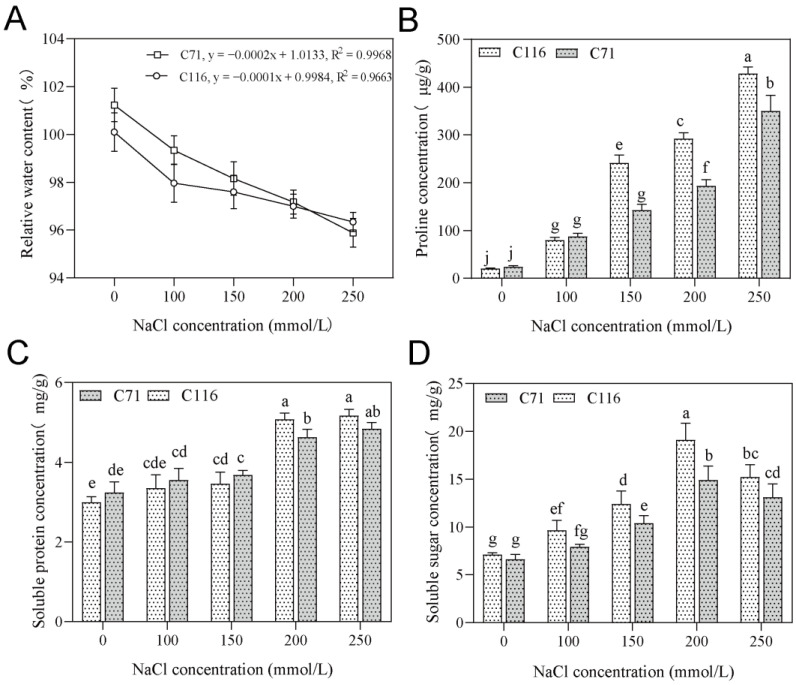
Accounting for variations in osmoregulatory substances under different salt concentrations. (**A**) The trend lines for RWC in both materials under salt stress: linear regression, coefficient of determination (*R*^2^); *y* and *x* are the vertical and horizontal values, respectively. Accumulation of proline (**B**), soluble protein (**C**), and soluble sugar (**D**) in two materials under different salt concentrations. The line bars represent the standard deviation of the means. Different letters denote significant differences at the *p* < 0.05 level found by MRT.

**Figure 3 genes-13-01904-f003:**
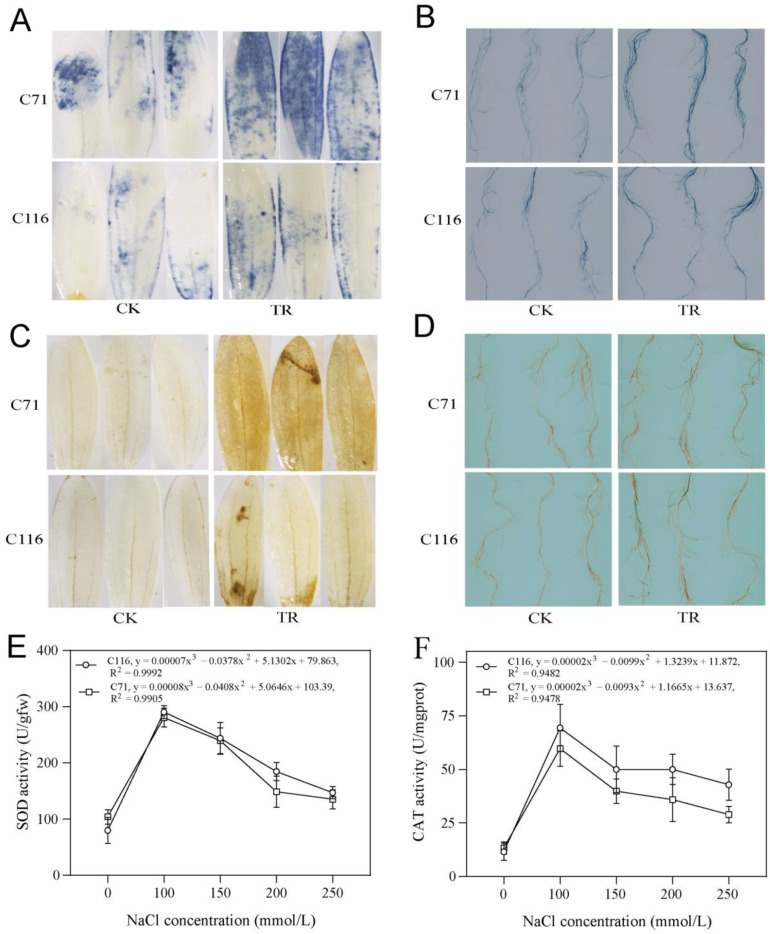
Antioxidant properties of resistant and sensitive materials under salt stress. (**A**) Flax leaves and (**B**) roots were chosen for NBT staining. (**C**) The cotyledons and (**D**) root systems of flax were used for DAB staining. CK, the control from 0 mmol/L NaCl; TR, the treatment from 200 mmol/L NaCl. (**E**) SOD enzyme activity assay and (**F**) CAT enzyme activity assay under different salt concentrations. In the curvilinear regression, *y* and *x* are the vertical and horizontal values, respectively, and *R*^2^ represents the coefficient of determination.

**Figure 4 genes-13-01904-f004:**
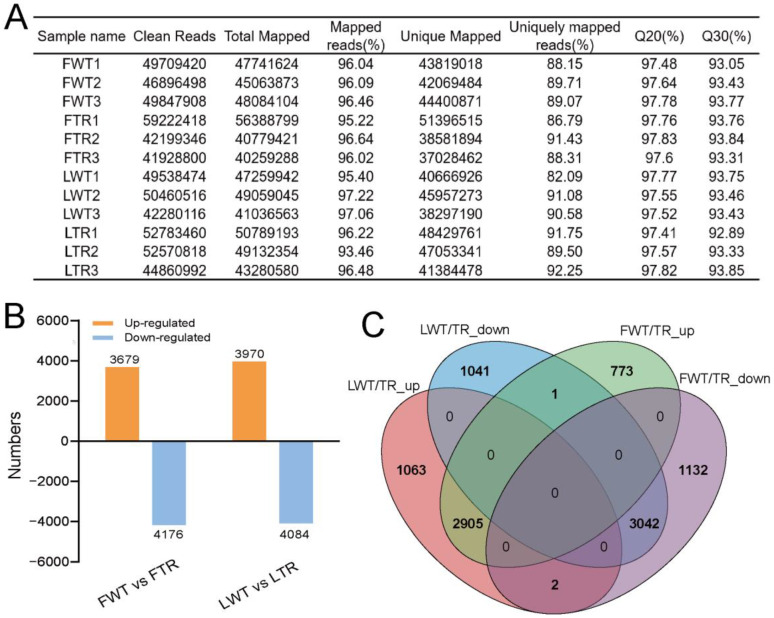
Statistical summary of flax transcriptomes. (**A**) Digest of data generated in flax transcriptome sequencing. (**B**) The number of upregulated (orange) and downregulated (blue) DEGs between each comparison. (**C**) Venn diagram showing the number of shared upregulated and downregulated DEGs among the two varieties. FWT, water treatment Fanni; FTR, NaCl stress-treated Fanni; LWT, water treatment Longya10; LTR, NaCl stress-treated Longya10.

**Figure 5 genes-13-01904-f005:**
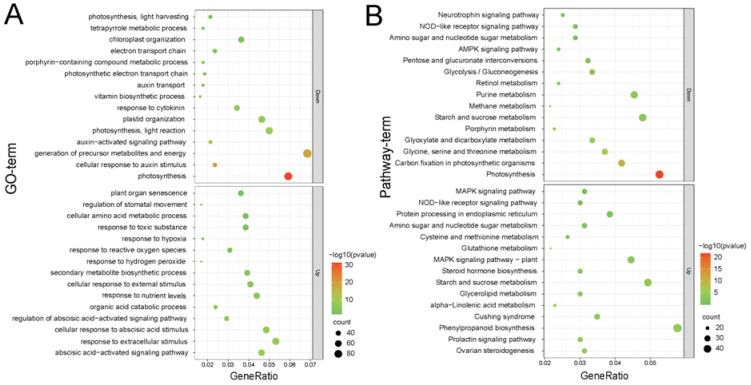
Gene enrichment of DEGs. (**A**) GO and (**B**) KEGG pathway enrichment analysis of upregulated DEGs and downregulated DEGs. Enrichment bubble plots were drawn by picking the top 15 terms for up- and downregulated DEGs.

**Figure 6 genes-13-01904-f006:**
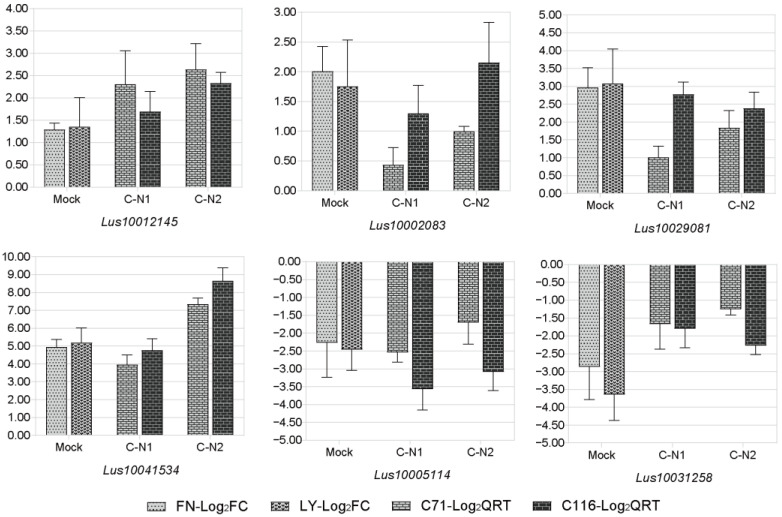
Validation of the RNA-seq data expression profile with qRT-PCR. The relative expression levels of six DEGs were calculated according to the 2^−ΔΔCt^ method using the *Actin* gene as an internal reference gene. The *x*-axis indicates the change of transcriptome data (Mock) and qRT-PCR data after exposure to 200 mM NaCl solution for 24 h (C-N1) and 72 h (C-N2). FN-Log_2_FC, Fanni log_2_ (fold change); LY-Log_2_FC, Longya10 log_2_ (fold change); C71-Log_2_QRT, C71 log_2_ (2^−ΔΔCt^); C116-Log_2_QRT, C116 log_2_ (2^−ΔΔCt^).

## Data Availability

All sequencing data generated for this study can be found in the NCBI database under accession number PRJNA860005 (https://www.ncbi.nlm.nih.gov/bioproject/PRJNA860005, accessed on 16 October 2022).

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
