# Peer review of "Adaptive Response and Transcriptomic Analysis of Flax (*Linum usitatissimum* L.) Seedlings to Salt Stress"

_genes, 2022, doi:10.3390/genes13101904_

Round 1

Reviewer 1 Report

Dear Authors

Reviewer suggestion: Minor revision

Overall, the manuscript provides a nice study in Adaptive response and transcriptomic analysis of flax (Linum 2 usitatissimum L.) seedlings to salt stress. However, the reviewer thinks there are some aspects that can be improved answered  as listed below:

1.    The introduction could be improved and more focused.

2.    This reviewer thinks that an additional section in relation to novel aspects of this work.

3. The discussion section needs to be improved.

4. The novelty of this research needs to improve and write clearly.

5. In introduction add some new references. add below references in Introduction:

-Encapsulation of Plant Biocontrol Bacteria with Alginate as a Main Polymer Material

-Salinity Stress: Toward Sustainable Plant Strategies and Using Plant Growth-Promoting Rhizobacteria Encapsulation for Reducing It

-Biopolymers for Biological Control of Plant Pathogens: Advances in Microencapsulation of Beneficial Microorganisms

Finally, I suggest to do a minor revision. 

Author Response

Point 1: The introduction could be improved and more focused.

Response 1: Dear reviewer, thank you again for your positive comments and valuable suggestions to improve the quality of our manuscript. In the revised manuscript, the second paragraph of this section has been reorganized and revised for the overall coherence of the introduction. The changes of our manuscript are highlighted with red colored.

Point 2: This reviewer thinks that an additional section in relation to novel aspects of this work.

Response 2: Dear reviewer, we are grateful for the valuable suggestion. As you are concerned, our transcriptome analysis should be noticed. In the revised manuscript, we have added the novelty of transcriptome analysis in the discussion.

Point 3: The discussion section needs to be improved.

Response 3: Dear reviewer, thanks for your suggestion. Based on the suggestions, we have made an extensive modification in the revised manuscript. The changes of our manuscript are highlighted with red colored.

Point 4: The novelty of this research needs to improve and write clearly.

Response 4: Dear reviewer, thanks for your professional review work on our article. In the resubmitted manuscript, our paper had been edited for English language at Wiley Editing Services.

Point 5: In introduction add some new references. add below references in Introduction:

Response 5: Dear reviewer, thank you for your recommendations. I benefited a lot from reading these papers you recommended. And according to your suggestion, I have added some new references in introduction.

Reviewer 2 Report

Manuscript "Adaptive response and transcriptomic analysis of flax (Linum usitatissimum L.) seedlings to salt stress" is very interesting.

General comments:

Authors investigated the differences in osmoregulator concentration and antioxidant capacity of resistant and susceptible materials at the seedling stage, and combined with transcriptomic data to elucidate the relationships between physiological response and gene expression under salt resistance in flax.
Obtained results were beneficial for breeders to screening salt-tolerant varieties in flax.
The results showed that plant height and root length of flax were inhibited, with C116 showed lower growth compared to C71. The concentrations of osmotic adjustment substances such as soluble sugars, soluble proteins and proline were higher in the resistant materials C116 than in the sensitive materials C71, under different concentrations of salt stress. C116 showed better rapid scavenging ability of ROS and maintained higher activities of antioxidant enzymes to balance salt injury stress by inhibiting growth under salt stress.

Detailed comments:

Introduction is perfect.
Description of plant material is very good. Unfortunately lack of descrpition of statisical analysis.
Lines 201-205: "2.6. Statistical analysis". This is the list the used programmes, and not the description of applied methods.
Lack information about distribution of observed traits.
Figure 2A needs regression models and coefficients of determination.
Figures 3E and 3F need regression models and coefficients of determination.

Need correction:

Line 175: "p-value" not "P-value"
L175: "log2"? This is incorrect notation.
L361: "log2"?
L362: "log2"?

Table S2: Lack information about presented values. Means? Table needs standard deviations.
Table S3: Lack information about presented values. Means? Table needs standard deviations.
Table S4: Phenotypic or genotypic correaltions?

Paper needs major revision.

Round 2

Reviewer 2 Report

Now, all is ok.